# Sparse Probabilistic Circuits via Pruning and Growing

**Meihua Dang**
CS Department
UCLA
mhdang@cs.ucla.edu

**Anji Liu**
CS Department
UCLA
liuanji@cs.ucla.edu

**Guy Van den Broeck**
CS Department
UCLA
guyvdb@cs.ucla.edu

## Abstract

Probabilistic circuits (PCs) are a tractable representation of probability distributions allowing for exact and efficient computation of likelihoods and marginals. There has been significant recent progress on improving the scale and expressiveness of PCs. However, PC training performance plateaus as model size increases. We discover that most capacity in existing large PC structures is wasted: fully-connected parameter layers are only sparsely used. We propose two operations: *pruning* and *growing*, that exploit the sparsity of PC structures. Specifically, the pruning operation removes unimportant sub-networks of the PC for model compression and comes with theoretical guarantees. The growing operation increases model capacity by increasing the size of the latent space. By alternatingly applying pruning and growing, we increase the capacity that is meaningfully used, allowing us to significantly scale up PC learning. Empirically, our learner achieves state-of-the-art likelihoods on MNIST-family image datasets and on Penn Tree Bank language data compared to other PC learners and less tractable deep generative models such as flow-based models and variational autoencoders (VAEs).

## 1   Introduction

Probabilistic circuits (PCs) [44, 3] are a unifying framework to abstract from a multitude of tractable probabilistic models. The key property that separates PCs from other deep generative models such as flow-based models [31] and VAEs [19] is their *tractability*. It enables them to compute various queries, including marginal probabilities, exactly and efficiently [45]. Therefore, PCs are increasingly used in inference-demanding applications such as enforcing algorithmic fairness [2, 4], making predictions under missing data [6, 18, 23], data compression [26], and anomaly detection [13].

Recent advancements in PC learning and regularization [40, 25], and efficient implementations [33, 30, 8] have been pushing the limits of PC's expressiveness and scalability such that they can even match the performance of less

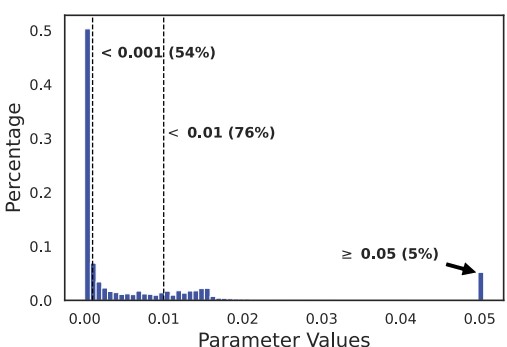

Figure 1: Histogram of parameter values for a state-of-the-art PC with 2.18M parameters on MNIST. 95% of the parameters have close-to-zero values.

tractable deep generative models, including flow-based models and VAEs. However, the performance of PCs plateaus as model size increases. This suggests that to further boost the performance of PCs, simply scaling up the model size does not suffice and we need to better utilize the available capacity.

36th Conference on Neural Information Processing Systems (NeurIPS 2022).

We discover that this might be caused by the fact that the capacity of large PCs is wasted. As shown in Figure 1, most parameters in a PC with 2.18M parameters have close-to-zero values, which have little effect on the PC distribution. Since existing PC structures usually have fully-connected parameter layers [25, 36], this indicates that the parameter values are only sparsely used.

In this work, we propose to better exploit the sparsity of large PC models by two structure learning primitives — *pruning* and *growing*. Specifically, the goal of the pruning operation is to identify and remove unimportant sub-networks of a PC. This is done by quantifying the importance of PC parameters w.r.t. a dataset using *circuit flows*, a theoretically-grounded metric that upper bounds the drop of log-likelihood caused by pruning. Compared to L1 regularization, the proposed pruning operator is more informed by the PC semantics, and hence quantifies the global effects of pruning much more effectively. Empirically, the proposed pruning method achieves a compression rate of 80-98% with at most 1% drop in likelihood on various PCs.

The proposed growing operation increases the model size by copying its existing components and injecting noise. In particular, when applied to PCs compressed by the pruning operation, growing produces larger PCs that can be optimized to achieve better performance. Applying pruning and growing iteratively can greatly refine the structure and parameters of a PC. Empirically, the log-likelihoods metric can improve by 2% to 10% after a few iterations. Compared to existing PC learners as well as less tractable deep generative models such as VAEs and flow-based models, our proposed method achieves state-of-the-art density estimation results on image datasets including MNIST, EMNIST, FashionMNIST, and the Penn Tree Bank language modeling task.[1]

## 2 Probabilistic Circuits

*Probabilistic circuits (PCs)* [44, 3] model probability distributions with a structured computation graph. They are an umbrella term for a large family of tractable probabilistic models including arithmetic circuits [9, 10], sum-product networks (SPNs) [35], cutset networks [36], and-or search spaces [28], and probabilistic sentential decision diagrams [21]. The syntax and semantics of PCs are defined as follows.

**Definition 1** (Probabilistic Circuit). A PC $\mathcal{C} := (\mathcal{G}, \boldsymbol{\theta})$ represents a joint probability distribution $p(\mathbf{X})$ over random variables $\mathbf{X}$ through a directed acyclic (computation) graph (DAG) $\mathcal{G}$ parameterized by $\boldsymbol{\theta}$. Similar to neural networks, each node in the DAG defines a computational unit. Specifically, the DAG $\mathcal{G}$ consists of three types of units — *input*, *sum*, and *product*. Every leaf node in $\mathcal{G}$ is an input unit; every inner unit $n$ (i.e., sum or product) receives *inputs* from its children $\mathsf{in}(n)$, and computes *output*, which encodes a probability distribution $p_n$ defined recursively as follows:

$$p_n(\boldsymbol{x}) := \begin{cases} f_n(\boldsymbol{x}) & \text{if } n \text{ is an input unit,} \\ \prod_{c \in \mathsf{in}(n)} p_c(\boldsymbol{x}) & \text{if } n \text{ is a product unit,} \\ \sum_{c \in \mathsf{in}(n)} \theta_{c|n} \cdot p_c(\boldsymbol{x}) & \text{if } n \text{ is a sum unit,} \end{cases} \tag{1}$$

where $f_n(\boldsymbol{x})$ is a univariate input distribution (e.g, Gaussian, Categorical), and $\theta_{c|n}$ denotes the parameter that corresponds to edge $(n, c)$ in the DAG. For every sum unit $n$, its input parameters sum up to one, i.e., $\sum_{c \in \mathsf{in}(n)} \theta_{c|n} = 1$. Intuitively, a product unit defines a factorized distribution over its inputs, and a sum unit represents a mixture over its input distributions with weights $\{\theta_{c|n} : c \in \mathsf{in}(n)\}$. Finally, the probability distribution of a PC (i.e., $p_{\mathcal{C}}$) is defined as the distribution represented by its root unit $r$ (i.e., $p_r(\boldsymbol{x})$), that is, its output neuron. The size of a PC, denoted $|\mathcal{C}| = |\boldsymbol{\theta}|$, is the number of parameters in $\mathcal{C}$. We assume w.l.o.g. that a PC alternates between layers of sum and product units before reaching its inputs. Figure 2 shows an example of a PC.

Computing the (log)likelihood of a PC $\mathcal{C}$ given a sample $\boldsymbol{x}$ is equivalent to evaluating its computation units in $\mathcal{G}$ in a feedforward manner following Equation 1. The key property that separates PCs from other deep probabilistic models such as flows [14] and VAEs [19] is their *tractability*, which is the ability to exactly and efficiently answer various probabilistic queries. This paper focuses on PCs that support linear time (w.r.t. model size) marginal probability computation, as they are increasingly used in downstream applications such as data compression [26] and making predictions under missing data [18], and also achieve on-par expressiveness [26, 25, 24]. To support efficient marginal inference, PCs need to be *smooth* and *decomposable*.

---

[1] Code and experiments are available at `https://github.com/UCLA-StarAI/SparsePC`.

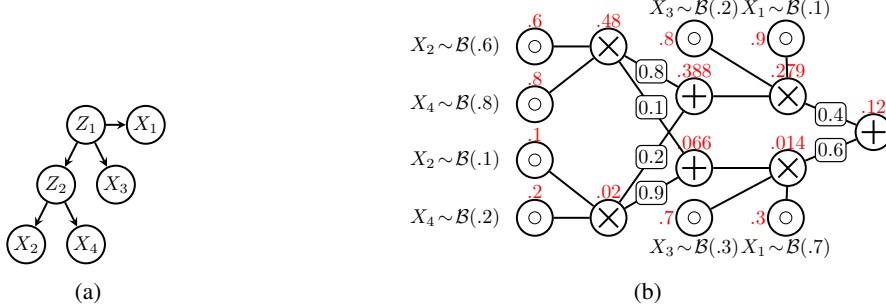

(a)           (b)

Figure 2: A smooth and decomposable PC (b) and an equivalent Bayesian network (a). The Bayesian network is over 4 variables $\mathbf{X} = \{X_1, X_2, X_3, X_4\}$ and 2 hidden variables $\mathbf{Z} = \{Z_1, Z_2\}$ with $h = 2$ hidden states. The feedforward computation order is from left to right; $\odot$ are input Bernoulli distributions, $\otimes$ are product units, and $\oplus$ are sum units; parameter values are annotated in the box. The probability of each unit given input assignment $\{X_1\!=\!0, X_2\!=\!1, X_3\!=\!0, X_4\!=\!1\}$ is labeled red.

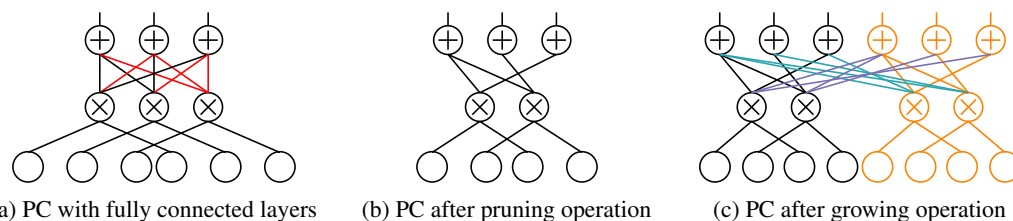

(a) PC with fully connected layers  (b) PC after pruning operation  (c) PC after growing operation

Figure 3: A demonstration of the pruning and growing operation. From 3a to 3b, the red edges are pruned. From 3b to 3c, the nodes are doubled, and each parameter is copied 3 times.

**Definition 2** (Smoothness and Decomposability [11]). The *scope* $\phi(n)$ of a PC unit $n$ is the set of input variables that it depends on; then, (1) a product unit is *decomposable* if its children have disjoint scope; (2) a sum unit is *smooth* if its children have identical scope. A PC is decomposable if all of its product units are decomposable; a PC is smooth if all of its sum units are smooth.

Decomposability ensures that every product unit encodes a well-defined factorized distribution over disjoint sets of variables; smoothness ensures that the mixture components of every sum units are well-defined over the same set of variables. Both structural properties will be the key to guaranteeing the effectiveness of the structure learning algorithms proposed in the following sections.

## 3 Probabilistic Circuit Model Compression via Pruning

Figure 1 shows that most parameters in a large PC are very close to zero. Given that these parameters are weights associated with mixture (sum unit) components, the corresponding edges and sub-circuits have little impact on the sum unit output. Hence, by pruning away these unimportant components, it is possible to significantly reduce model size while retaining model expressiveness. Figure 3b illustrates the result of pruning five (red) edges from the PC in Figure 3a. Given a PC and a dataset, our goal is to efficiently identify a set of edges to prune, such that the log-likelihood gap between the pruned PC and the original PC on the given dataset is minimized.

**Pruning by parameters.** The parameter value statistics in Figure 1 suggest that a natural criterion is to prune edges by the magnitude of their corresponding parameter. This leads to the EPARAM (edge parameters) heuristic, which selects the set of edges with the smallest parameters. However, edge parameters themselves are insufficient to quantify the importance of inputs to a sum unit in the entire PC's distribution. The parameters of a sum unit are normalized to be 1 so they only contain local information about the mixture components. Specifically, $\theta_{c|n}$ merely defines the relative importance of edge $(n, c)$ in the conditional distribution represented by its corresponding sum unit $n$, not the joint distribution of the entire PC. Figure 4a illustrates what happens when the edge with the smallest parameter is pruned from the PC in Figure 2.

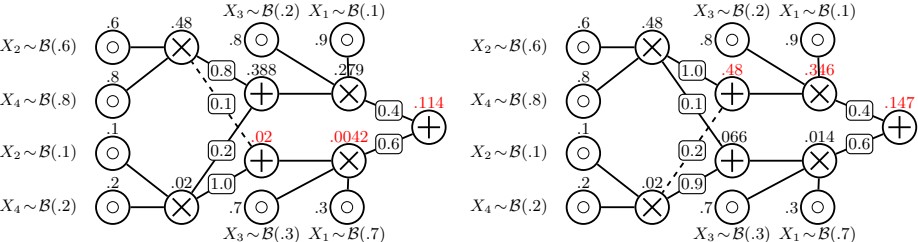

(a) EPARAM removes the edge with $\theta = 0.1$        (b) EFLOW removes the edge with $\theta = 0.2$

Figure 4: A case study comparing pruning heuristics (EPARAM and EFLOW) on the PC in Figure 2 given sample $\{X_1 = 0, X_2 = 1, X_3 = 0, X_4 = 1\}$. The pruned edges are dashed and parameters are re-normalized. Compared to the likelihood computed in Figure 2, the changed likelihoods are in red, showing that pruning by flows results in less likelihood decrease.

---

**Algorithm 1:** PC sampling

---

**Input** : a PC representing joint probability $p_\mathcal{C}(\mathbf{X})$
**Output** : an instance $\boldsymbol{x}$ sampled from $p_\mathcal{C}$

1 **Function** SAMPLE(n)
2     **if** *n is a an input unit* **then**
3        $f_n(X) \leftarrow$ univariate distribution of $n$; **return** sample $x \sim f_n(X)$
4     **else if** *n is a product unit* **then**
5        $\boldsymbol{x}_c \leftarrow$ SAMPLE(c) foreach $c \in \mathsf{in}(n)$; **return** Concatenate($\{\boldsymbol{x}_c\}_{c \in \mathsf{in}(n)}$)
6     **else** $n$ is a sum unit
7        sample an input $c^*$ proportional to $\{\theta_{c|n}\}_{c \in \mathsf{in}(n)}$; **return** SAMPLE($c^*$)
8 **return** SAMPLE(r) where $r$ is the root of PC $\mathcal{C}$

---

However, as shown in Figure 4b, pruning another edge delivers better likelihoods as it accounts more for the "global influence" of edges on the PC's output. This global influence is highly related to the probabilistic "circuit flow" semantics of PCs. We will introduce circuit flows later in this section, along with their corresponding heuristics EFLOW. Before that, we first introduce an intermediate concept based on the notion of generative significance of PCs.

**Pruning by generative significance.** A more informed pruning strategy needs to consider the global impact of edges on the distribution represented by the output of the PC. To achieve this, instead of viewing the distribution $p_\mathcal{C}$ in a feedforward manner following Equation 1, we quantify the significance of a unit or edge by the probability that it will be "activated" when drawing samples from the PC. Indeed, if the presence of an edge is hardly ever relevant to the generative sampling process, removing it will not significantly affect the PC's distribution.

Algorithm 1 shows how to draw samples from a PC distribution through a recursive implementation: (1) for an input unit $n$ defined on variable $X$ (line 3), the algorithm randomly samples value $x$ according to its input univariate distribution; (2) for a product unit (line 5), by decomposability its children have disjoint scope, thus we draw samples from all input units and then concatenate the samples together; (3) for a sum unit $n$ (line 7), by smoothness its children have identical scope, thus we first randomly sample one of its input units according to the categorical distribution defined by sum parameters $\{\theta_{c|n} : c \in \mathsf{in}(n)\}$, and then sample from this input unit recursively. Besides actually drawing samples from the PC, we can also compute the probability that $n$ will be visited during the sampling process. This provides a good measure of the importance of unit $n$ to the PC distribution as a whole, which we define as the *top-down probability*.

**Definition 3** (Top-down Probability). The top-down probability of each unit $n$ in a PC with parameters $\boldsymbol{\theta}$ is defined recursively as follows, assuming alternating sum and product layers:

$$q(n; \boldsymbol{\theta}) := \begin{cases} 1 & \text{if } n \text{ is the root unit,} \\ \sum_{m \in \mathsf{out}(n)} q(m; \boldsymbol{\theta}) & \text{if } n \text{ is a sum unit,} \\ \sum_{m \in \mathsf{out}(n)} \theta_{n|m} \cdot q(m; \boldsymbol{\theta}) & \text{if } n \text{ is a product unit,} \end{cases}$$

where $\mathsf{out}(n)$ are the units that take $n$ as input in the feedforward computation. Moreover, the top-down probability of a sum edge $(n, c)$ is defined as $q(n, c; \boldsymbol{\theta}) = \theta_{c|n} \cdot q(n; \boldsymbol{\theta})$.

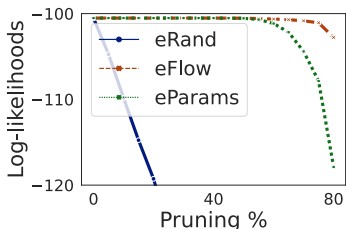
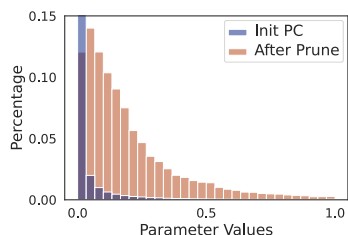

(a) Comparison of heuristics ERAND, EPARAM, and EFLOW. Heuristic EFLOW can prune up to 80% of the parameters without much loglikelihoods decrease.

(b) Histogram of parameters before (the same as in Figure 1) and after pruning. The parameter values take higher significance after pruning.

Figure 5: Empirical evaluation of the pruning operation.

The top-down probability of the root is always 1; a product unit passes its top-down probability to all its inputs, and a sum unit distributes its top-down probability to its inputs proportional to the corresponding edge weights. Therefore, the top-down probability of a non-root unit is summing over all probabilities it receives from its outputs.

The top-down probability of all PC units and sum edges can be computed in a single backward pass over the PC's computation graph. Following the intuition that the top-down probability defines the probability that units will be visited during the sampling process, pruning edges with the smallest top-down probability constitutes a reasonable pruning strategy.

**Pruning by circuit flows.** The top-down probability $q(n; \boldsymbol{\theta})$ represents the probability of reaching unit $n$ in an unconditional random sampling process. Despite its ability to capture global information of PC parameters, the top-down probability is not tailored to a specific dataset. Therefore, to further utilize the dataset information, we can measure the probability of reaching certain units/edges in the sampling process *conditioning on some instance $\boldsymbol{x}$ being sampled*. To bridge this gap, we define circuit flow as a sample-dependent version of the top-down probability.

**Definition 4** (Circuit Flow[2])**.** For a given PC with parameters $\boldsymbol{\theta}$ and example $\boldsymbol{x}$, the circuit flow of unit $n$ on example $\boldsymbol{x}$ is the probability that $n$ will be visited during the sampling procedure conditioned on $\boldsymbol{x}$ being sampled. This can be computed recursively as follows, assuming alternating sum and product layers:

$$
F_n(\boldsymbol{x}) = \begin{cases} 1 & \text{if } n \text{ is the root unit,} \\ \sum_{m \in \mathsf{out}(n)} F_m(\boldsymbol{x}) & \text{if } n \text{ is a sum unit,} \\ \sum_{m \in \mathsf{out}(n)} \frac{\theta_{n|m} \cdot p_n(\boldsymbol{x})}{p_m(\boldsymbol{x})} \cdot F_m(\boldsymbol{x}) & \text{if } n \text{ is a product unit.} \end{cases}
$$

Similarly, the edge flow $F_{n,c}(\boldsymbol{x})$ on sample $\boldsymbol{x}$ is defined by $F_{n,c}(\boldsymbol{x}) = \theta_{c|n} \cdot p_c(\boldsymbol{x})/p_n(\boldsymbol{x}) \cdot F_n(\boldsymbol{x})$. We further define $F_{n,c}(\mathcal{D}) = \sum_{\boldsymbol{x} \in \mathcal{D}} F_{n,c}(\boldsymbol{x})$ as the *aggregate edge flow* over dataset $\mathcal{D}$.

Effectively, we can think of $\theta_{n|m}^{\boldsymbol{x}} := \theta_{n|m} \cdot p_n(\boldsymbol{x})/p_m(\boldsymbol{x})$ as the posterior probability of component $n$ in the mixture of sum unit $m$ *conditioned on observing sample $\boldsymbol{x}$*. Then, circuit flow is the top-down probability under this $\boldsymbol{\theta}^{\boldsymbol{x}}$ reparameterization of the circuit: $F_n(\boldsymbol{x}) = q(n; \boldsymbol{\theta}^{\boldsymbol{x}})$ and $F_{n,c}(\boldsymbol{x}) = q(n, c; \boldsymbol{\theta}^{\boldsymbol{x}})$.

Circuit flow $F_n(\boldsymbol{x})$ defines the probability of reaching unit $n$ in the top-down sampling procedure of Algorithm 1, given that the sampled instance is $\boldsymbol{x}$. Therefore, edge flow $F_{n,c}(\boldsymbol{x})$ is a natural metric of the importance of edge $(n, c)$ given $\boldsymbol{x}$. Intuitively, the aggregate circuit flow measures how many expected samples "flow" through certain edges. We write EFLOW to refer to the heuristic that prunes edges with the smallest aggregate circuit flow.

**Empirical Analysis.** Figure 5a compares the effect of pruning heuristics EPARAM, EFLOW, as well as an uninformed strategy, prune randomly, which we denote as ERAND. It shows that both EPARAM and EFLOW are reasonable pruning strategy, however, as we increase the percentage of

---

[2]Earlier work defined "circuit flow" or "expected circuit flow" in the context of parameter learning [4, 25, 7], without observing the connection to sampling. We contribute its more intuitive sampling semantics here.

pruned parameters, EFLOW has less log-likelihoods drop compared with EPARAM. Using EFLOW heuristics we can pruning up to 80% of the parameters without much log-likelihoods drop. As shown in Figure 5b, the parameter distribution is more balanced after pruning compared to Figure 1, indicating a higher significance of each edge. Section 6 will provide more empirical results. Before that, we first theoretically verify the effectiveness of the EFLOW heuristic in the next section.

## 4 Bounding and Approximating the Loss of Likelihood

In this section, we theoretically quantify the impact of edge pruning on model performance. In particular, we establish an upper bound on the log-likelihood drop $\Delta \mathcal{LL}$ on a given dataset $\mathcal{D}$ by comparing (i) the original PC $\mathcal{C}$ and (ii) the pruned PC $\mathcal{C}_{\setminus \mathcal{E}}$ caused by pruning away edges $\mathcal{E}$:

$$\Delta \mathcal{LL}(\mathcal{D}, \mathcal{C}, \mathcal{E}) = \mathcal{LL}(\mathcal{D}, \mathcal{C}) - \mathcal{LL}(\mathcal{D}, \mathcal{C}_{\setminus \mathcal{E}}). \tag{2}$$

We start from the case of pruning one edge (i.e., $|\mathcal{E}| = 1$ in Equation 2). In this case, the loss of likelihood can be quantified exactly using flows and edge parameters:

**Theorem 1** (Log-likelihood drop of pruning one edge). *For a PC $\mathcal{C}$ and a dataset $\mathcal{D}$, the loss of log-likelihood by pruning away edge $(n, c)$ is*

$$\Delta \mathcal{LL}(\mathcal{D}, \mathcal{C}, \{(n,c)\}) = \frac{1}{|\mathcal{D}|} \sum_{\boldsymbol{x} \in \mathcal{D}} \log\left(\frac{1 - \theta_{c|n}}{1 - \theta_{c|n} + \theta_{c|n} \, \mathrm{F}_n(\boldsymbol{x}) - \mathrm{F}_{n,c}(\boldsymbol{x})}\right) \le \frac{-1}{|\mathcal{D}|} \sum_{\boldsymbol{x} \in \mathcal{D}} \log(1 - \mathrm{F}_{n,c}(\boldsymbol{x})).$$

See proof in Appendix B.1. By computing the second term in Theorem 1, we can pick the edge with the smallest log-likelihood drop. Additionally, the third term characterizes the log-likelihood drop without re-normalizing parameters of $\theta_{\cdot|n}$. It suggests pruning the edge with the smallest edge flow. A key insight from Theorem 1 is that the log-likelihood drop depends explicitly on the edge flow $\mathrm{F}_{n,c}(\boldsymbol{x})$ and unit flow $\mathrm{F}_n(\boldsymbol{x})$. This matches the intuition from Section 3 and suggests that the circuit flow heuristic proposed in the previous section is a good approximation of the derived upper bound.

Next, we bound the log-likelihood drop of pruning multiple edges.

**Theorem 2** (Log-likelihood drop of pruning multiple edges). *Let $\mathcal{C}$ be a PC and $\mathcal{D}$ be a dataset. For any set of edges $\mathcal{E}$ in $\mathcal{C}$, if $\forall \boldsymbol{x} \in \mathcal{D}, \sum_{(n,c) \in \mathcal{E}} \mathrm{F}_{n,c}(\boldsymbol{x}) < 1$, the log-likelihood drop by pruning away $\mathcal{E}$ is bounded and approximated by*

$$\Delta \mathcal{LL}(\mathcal{D}, \mathcal{C}, \mathcal{E}) \le -\frac{1}{|\mathcal{D}|} \sum_{\boldsymbol{x}} \log\left(1 - \sum_{(n,c) \in \mathcal{E}} \mathrm{F}_{n,c}(\boldsymbol{x})\right) \approx \frac{1}{|\mathcal{D}|} \sum_{(n,c) \in \mathcal{E}} \mathrm{F}_{n,c}(\mathcal{D}). \tag{3}$$

Proof of this theorem is provided in Appendix B.2. We first look at the second term of Equation 3. Although it provides an upper bound to the performance drop, it cannot be used as a pruning heuristic since the bound does not decompose over edges. And hence finding the set of edges with the lowest score requires evaluating the bound exponentially many times with respect to the number of pruned edges. Therefore, we do an additional approximation step of the bound via Taylor expansion, which leads to the third term of Equation 3. This approximation matches the EFLOW heuristic by a constant factor $1/|\mathcal{D}|$, which theoretically justifies the effectiveness of the heuristic. Figure 6 empirically compares the actual log-likelihood drop and the quantity computed from the circuit flow heuristic (that is, the approximate upper bound) for different percentages of pruned parameters. We see that the approximate bound matches closely to the actual log-likelihood drop.

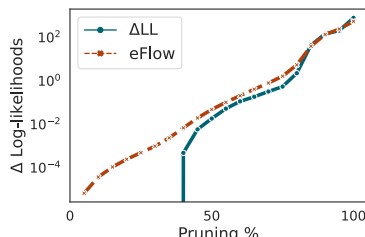

Figure 6: Comparing the actual loglikelihood drop ($\Delta$LL) and EFLOW heuristics (the approximated upper bound in Equation 3). The approximate bound matches closely to the actual loglikelihood drop.

## 5 Scalable Structure Learning

The pruning operator improves two aspects of PCs. First, as shown in Figure 5b, model parameters are more balanced after pruning. Second, pruning removes sub-circuits with negligible contributions

to the model's distribution. If we treat PCs as hierarchical mixtures of components, pruning can be regarded as an implicit structure learning step that removes the "unimportant" components for each mixture. However, since pruning only decreases model capacity, it is impossible to get a more expressive PC than the original one. To mitigate this problem, we propose a *growing* operation to increase the capacity of a PC by introducing more components for each mixture. Pruning and growing together define a scalable structure learning algorithm for PCs.

**Growing.** *Growing* is an operator that increases model size by copying its existing components and injecting noise. As shown in Figure 3, after applying the growing operation on the original PC in Figure 3b, we can get a new grown PC as in Figure 7. Specifically, the growing operation is applied to units, edges, and parameters respectively: (1) for units, growing operates on every PC unit $n$ and creates another copy $n^{\text{new}}$; (2) for edges, the sum edge $(n, c)$ from the original PC (Figure 3b) are copied three times to the grown PC (Figure 7): from new parent to new child $(n^{\text{new}}, c^{\text{new}})$, from old parent to new child $(n, c^{\text{new}})$, and from new parent to old child $(n^{\text{new}}, c)$; product edges are added to connect the copied version of a product unit and its copied inputs; (3) a new parameter $\theta_{c|n}^{\text{new}}$ is a noisy

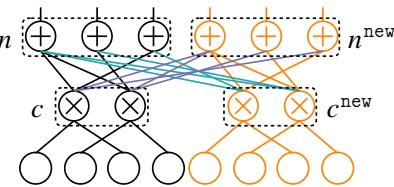

Figure 7: Growing operation. Each unit is doubled, and each parameterized edge is copied 3 times: $(n^{\text{new}}, c^{\text{new}})$ (orange), $(n^{\text{new}}, c)$ (purple), and $(n, c^{\text{new}})$ (green).

copy of an old parameter $\theta_{c|n}$, that is $\theta_{c|n}^{\text{new}} \leftarrow \epsilon \cdot \theta_{c|n}$ where $\epsilon \sim \mathcal{N}(1, \sigma^2)$ and $\sigma^2$ controls the Gaussian noise variance. Gaussian noise is added to the copied parameters to ensure that after we apply the growing operation, parameter learning algorithms can find diverse parameters for different copies. After a growing operation, the PC size is 4 times the original PC size. Algorithm 3 in appendix shows a feedforward implementation of the growing operation.

**Structure Learning through Pruning and Growing.** The proposed pruning and growing algorithms can be applied iteratively to refine the structure and parameters of an initial PC. Specifically, since the growing operator increases the number of PC parameters by a factor of 4, applying growing after pruning 75% of the edges from an initial PC keeps the number of parameters unchanged. We propose a joint structure and parameter learning algorithm for PCs that uses these two operations. Specifically, starting from an initial PC, we apply 75% pruning, growing, and parameter learning iteratively until convergence. We utilize HCLTs [25] as initial PC structure as it has the state-of-the-art likelihood performance. Note that this structure learning pipeline can be applied to any PC structure.

**Parameter Estimation.** We use a stochastic mini-batch version of Expectation-Maximization optimization [2]. Specifically, at each iteration, we draw a mini-batch of samples $\mathcal{D}_B$, compute aggregated circuit flows $\text{F}_{n,c}(\mathcal{D}_B)$ and $\text{F}_n(\mathcal{D}_B)$ of these samples (E-step), and then compute new parameter $\theta_{c|n}^{\text{new}} = \text{F}_{n,c}(\mathcal{D}_B)/\text{F}_n(\mathcal{D}_B)$. The parameters are then updated with learning rate $\alpha$: $\theta^{t+1} \leftarrow \alpha\theta^{\text{new}} + (1 - \alpha)\theta^t$ (M-step). Empirically this approach converges faster and is better regularized compared to full-batch EM.

**Parallel Computation.** Existing approaches to scaling up learning and inference with PCs, such as Einsum networks [33], utilize fully connected parametrized layers (Figure 3a) of PC structures such as HCLT [25] and RatSPN [34]. These structures can be easily vectorized to utilize deep learning packages such as PyTorch. However, the sparse structure learned by pruning and growing is not easily vectorized as a dense matrix operation. We therefore implement customized GPU kernels to parallelize the computation of parameter learning and inference based on Juice.jl [8], an open-source Julia package for learning PCs. The kernels segment PC units into layers such that the units in each layer are independent. Thus, the computation can be fully parallelized on the GPU. As a result, we can train PCs with millions of parameters in less than half an hour.

## 6  Experiments

We now evaluate our proposed method pruning and growing on two different sets of density estimation benchmarks: (1) the MNIST-family image generation datasets including MNIST [22], EMNIST [5], and FashionMNIST [46]; (2) the character-level Penn Tree Bank language modeling task [27].

Section 6.1 first reports the best results we get on image datasets and language modeling tasks via the structure learning procedure proposed in Section 5. Section 6.2 then shows the effect of pruning and growing operations via two detailed experimental settings. It studies two different constrained optimization problems: finding the smallest PC for a given likelihood via model compression and finding the best PC of a given size via structure learning.

**Settings.** For all experiments, we use hidden Chow-Liu Trees (HCLTs) [25] with the number of latent states in $\{16, 32, 64, 128\}$ as initial PC structures. We train the parameters of PCs with stochastic mini-batch EM (cf. Section 5). We perform early stopping and hyperparameter search using a validation set and report results on the test set. Please refer to Appendix C for more details. We use mean test set bits-per-dimension (bpd) as the evaluation criteria, where $\text{bpd}(\mathcal{D}, \mathcal{C}) = -\mathcal{LL}(\mathcal{D}, \mathcal{C})/(\log(2) \cdot m)$ and $m$ is the number of features in dataset $\mathcal{D}$.

## 6.1 Density Estimation Benchmarks

**Image Datasets.** The MNIST-family datasets contain gray-scale pixel images of size $28 \times 28$ where each pixel takes values in $[0, 255]$. We split out 5% of training data as a validation set. We compare with two competitive PC learning algorithms: HCLT [25] and RatSPN [34], one flow-based model: IDF [17], and three VAE-based methods: BitSwap [20], BB-ANS [41], and McBits [38]. For a fair comparison, we implement RatSPN structures ourselves and use the same training pipeline and EM optimizer as our proposed method. Note that EinsumNet [33] also uses RatSPN structures but with a PyTorch implementation so its comparison is subsumed by comparison with RatSPN. All 7 methods are tested on MNIST, 4 splits of EMNIST and FashionMNIST. As shown in Table 1, the best results are bold. We see that our proposed method significantly outperforms all other baselines on all datasets, and establishes new state-of-the-art results among PCs, flows, and VAE models. More experiment details are in Appendix C.

Table 1: Density estimation performance on MNIST-family datasets in test set bpd.

| Dataset | Sparse PC (ours) | HCLT | RatSPN | IDF | BitSwap | BB-ANS | McBits |
|---|---|---|---|---|---|---|---|
| MNIST | **1.14** | 1.20 | 1.67 | 1.90 | 1.27 | 1.39 | 1.98 |
| EMNIST(MNIST) | **1.52** | 1.77 | 2.56 | 2.07 | 1.88 | 2.04 | 2.19 |
| EMNIST(Letters) | **1.58** | 1.80 | 2.73 | 1.95 | 1.84 | 2.26 | 3.12 |
| EMNIST(Balanced) | **1.60** | 1.82 | 2.78 | 2.15 | 1.96 | 2.23 | 2.88 |
| EMNIST(ByClass) | **1.54** | 1.85 | 2.72 | 1.98 | 1.87 | 2.23 | 3.14 |
| FashionMNIST | **3.27** | 3.34 | 4.29 | 3.47 | 3.28 | 3.66 | 3.72 |

**Language Modeling Task.** We use the Penn Tree Bank dataset with standard processing from Mikolov et al. [29], which contains around 5M characters and a character-level vocabulary size of 50. The data is split into sentences with a maximum sequence length of 288. We compare with three competitive normalizing-flow-based models: Bipartite flow [42] and latent flows [48] including AF/SCF and IAF/SCF, since they are the only comparable work with non-autoregressive language modeling. As shown in Table 2, the proposed method outperforms all three baselines.

Table 2: Character-level language modeling results on Penn Tree Bank in test set bpd.

| Dataset | Sparse PC (ours) | Bipartite flow [42] | AF/SCF [48] | IAF/SCF [48] |
|---|---|---|---|---|
| Penn Tree Bank | **1.35** | 1.38 | 1.46 | 1.63 |

## 6.2 Evaluating Pruning and Growing

**What is the Smallest PC for the Same Likelihood?** We evaluate the ability of pruning based on circuit flows to do effective model compression by iteratively pruning a $k$-fraction of the PC parameters and then fine-tuning them until the final training log-likelihood does not decrease by more than 1%. Specifically, we take pruning percentage $k$ from $\{0.05, 0.1, 0.3\}$. As shown in Figure 8, we can achieve a compression rate of 80-98% with negligible performance loss on PCs. Besides, by

fixing the number of latent parameters (x-axis) and comparing bpp across different numbers of latent states (legend), we discover that compressing a large PC to a get smaller PC yields better likelihoods compared to directly training an HCLT with the same number of parameters from scratch. This can be explained by the sparsity of compressed PC structures, as well as a smarter way of finding good parameters: learning a better PC with larger size and compressing it down to a smaller one.

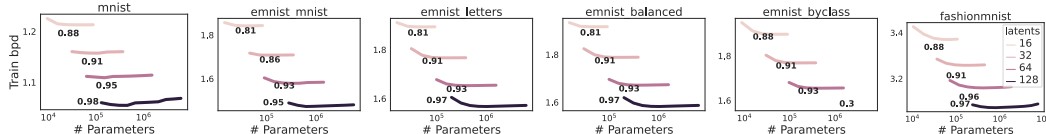

Figure 8: Model compression via pruning and finetuning. We report the training set bpd (y-axis) in terms of the number of parameters (x-axis) for different numbers of latent states. For each curve, compression starts from the right (initial PC #Params $|\mathcal{C}^{\texttt{init}}|$) and ends at the left (compressed PC #Params $|\mathcal{C}^{\texttt{com}}|$); compression rate $(1 - |\mathcal{C}^{\texttt{com}}| / |\mathcal{C}^{\texttt{init}}|)$ is annotated next to each curve.

**What is the Best PC for the Same Size?** We evaluate structure learning that combines pruning and growing as proposed in Section 5. Starting from an initial HCLT, we iteratively prune 75% of the parameters, grow again, and fine-tune until meeting the stopping criteria. As shown in Figure 9, our method consistently improve the likelihoods of initial PCs for different numbers of latent states among all datasets.

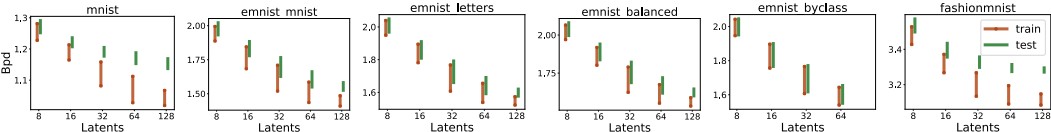

Figure 9: Structure learning via 75% pruning, growing and finetuning. We report bpd (y-axis) on both train (red) and test set (green) in terms of the number of latent states (x-axis). For each curve, training starts from the top (large bpd) and ends at the bottom (small bpd).

## 7 Related Work

Improving the expressiveness of PCs has been a central topic in the literature. Predominant works focus on the structure learning algorithms that iteratively modify PC structures to progressively fit the data [7, 24, 15]. Alternatively, a recent trend is to construct PCs with good initial structures and only perform parameter learning afterwards [36, 1, 47]. For example, EiNets [33], RAT-SPNs [34], and XPCs [12] use randomly generated structures, HCLTs [25] and ID-SPNs [37] define PCs dependent on the pairwise correlation on variables, and learn direct and indirect variable correlations among variables. There are also a few works that boost PC performance with the expressive power of neural networks. CSPNs [39] harness neural networks to learn expressive conditional density estimators, and HyperSPN [40] utilizes neural networks for better PC regularization.

Pruning and growing have been introduced in deep neural networks to exploit sparsity [16]. Similar strategies can be adopted in probabilistic circuits. Patil et al. [32] prune decision trees according to validation accuracy reduction. ResSPNs uses the lottery ticket hypothesis for weight pruning to gain more compact PCs [43]. However, the neural network pruning methods and above PC pruning methods mainly focus on pruning by parameters. In contrast to these, our work develops better pruning strategies based on semantic properties of PCs.

## 8 Conclusions

We propose structure learning of probabilistic circuits by combining pruning and growing operations to exploit the sparsity of PC structures. We show significant empirical improvements in the density estimation tasks of PCs compared to existing PC learners and competing flow-based models and VAEs. All our Sparse-PC learning and inference algorithms are available as GPU-parallel implementations.

**Acknowledgements** This work was funded in part by the DARPA Perceptually-enabled Task Guidance (PTG) Program under contract number HR00112220005, NSF grants #IIS-1943641, #IIS-1956441, #CCF-1837129, Samsung, CISCO, a Sloan Fellowship, and a UCLA Samueli Fellowship. We thank Honghua Zhang for proofreading and insightful comments on this paper's final version.

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
