# A Pseudocode

In this section, we list the detailed algorithms for pruning (Section 3), growing (Section 5), circuit flows computation (Definition 4), and mini-batch Expectation Maximization (Section 5).

Algorithm 2 shows how to prune $k$ percentage edges from PC $\mathcal{C}$ following heuristic $h$.

---

**Algorithm 2:** Prune($\mathcal{C}, h, k$)

---

**Input** : a non-deterministic PC $\mathcal{C}$, heuristic $h$ deciding which edge to prune, $h$ can be EFLOW, ERAND, or EPARAM, percentage of edges to prune $k$

**Output** : a PC $\mathcal{C}$' after pruned

1 old2new $\leftarrow$ mapping from input PC $n \in \mathcal{C}$ to pruned PC
2 $s(n, c) \leftarrow$ compute a score for each edge $(n, c)$ based on heuristic $h$
3 $f(n, c) \leftarrow false$
4 $f(n, c) \leftarrow true$ if $s(n, c)$ ranks the last $k$
5 // visit children before parents
6 **foreach** $n \in \mathcal{C}$ **do**
7     **if** $n$ *is a leaf* **then**
8         | old2new$[n] \leftarrow n$
9     **else if** $n$ *is a sum* **then**
10         | old2new$[n] \leftarrow \bigoplus([$old2new$(c)$ for $c \in$ in$(n)$ and if $f(n,c)])$
11     **else** $n$ is a product
12         | old2new$[n] \leftarrow \bigotimes([$old2new$(c)$ for $c \in$ in$(n)])$
13 **return** old2new$[n_r]$ where $n_r$ is the root of $\mathcal{C}$

---

Algorithm 3 shows show a feedforward implementation of growing operation.

---

**Algorithm 3:** Grow($\mathcal{C}, \sigma^2$)

---

**Input** : a PC $\mathcal{C}$, Gaussian noisy variance $\sigma^2$

**Output** : a PC $\mathcal{C}$' after growing operation

1 old2new $\leftarrow$ a dictionary mapping input PC units $n \in \mathcal{C}$ to units of the growed PC
2 **foreach** $n \in \mathcal{C}$ **do** // visit children before parents
3     **if** $n$ is an input unit **then** old2new$[n] \leftarrow (n, \mathsf{deepcopy}(n))$
4     **else**
5         chs_1, chs_2 $\leftarrow [$old2new$[c][0]$ for $c$ in in$(n)], [$old2new$[c][1]$ for $c$ in in$(n)]$
6         **if** $n$ is a product unit **then** old2new$[n] \leftarrow (\bigotimes(\mathsf{chs\_1}), \bigotimes(\mathsf{chs\_2}))$
7         **else if** $n$ *is a sum unit* **then**
8             | $n_1, n_2 \leftarrow \bigoplus([\mathsf{chs\_1}, \mathsf{chs\_2}]), \bigoplus([\mathsf{chs\_1}, \mathsf{chs\_2}])$
9             | $\boldsymbol{\theta}_{|n_i} \leftarrow \mathsf{normalize}([\boldsymbol{\theta}_{|n}, \boldsymbol{\theta}_{|n}]) \times \boldsymbol{\epsilon})$    $\epsilon \sim \mathcal{N}(\mathbf{1}, \sigma^2)$ for $i$ in $[1,2]$
10             | old2new$[n] \leftarrow (n_1, n_2)$
11 **return** old2new$[r][0]$ // $r$ is the root unit of $\mathcal{C}$

---

Algorithm 4 computes the circuit flows of a sample $\boldsymbol{x}$ given PC $\mathcal{C}$ with parameters $\boldsymbol{\theta}$ though one forward pass (line 1) and one backward pass (line 2-8).

---
**Algorithm 4:** CircuitFlow($\mathcal{C},\boldsymbol{\theta},\boldsymbol{x}$)
---
**Input** : a PC $\mathcal{C}$ with parameters $\boldsymbol{\theta}$; sample $\boldsymbol{x}$
**Output**: circuit flow flow$[n,c]$ for each edge $(n,c)$ and flow$[n]$ for each node $n$
1 $\forall n \in \mathcal{C}$, p$[n] \leftarrow p_n(\boldsymbol{x})$ computed as in Equation 1
2 For root $n_r$, flow$[n] \leftarrow 1$
3 **for** $n \in \mathcal{C}$ *in backward order* **do**
4     flow$[n] \leftarrow \sum_{g \in \text{out}(n)}$ flow$[g]$
5     **if** *n is a sum node* **then**
6         $\forall c \in \text{in}(n)$, flow$[n,c] \leftarrow \theta_{c|n} \frac{\text{p}[c]}{\text{p}[n]}$ flow$[n]$
7     **else**
8         $\forall c \in \text{in}(n)$, flow$[n,c] \leftarrow$ flow$[n]$
---

Algorithm 5 shows the pipeline of mini-batches Expectation Maximization algorithm given PC $\mathcal{C}$, dataset $\mathcal{D}$, batch size $B$ and learning rate $\alpha$.

---
**Algorithm 5:** StochasticEM($\mathcal{C},\mathcal{D};B,\alpha$)
---
**Input** : a PC $\mathcal{C}$; dataset $\mathcal{D}$; batch size $B$; learning rate $\alpha$
**Output**: parameters $\boldsymbol{\theta}$ estimated from $\mathcal{D}$
1 $\theta \leftarrow$ random initialization
2 For root $n_r$, flow$[n] \leftarrow 1$
3 **while** *not converged or early stopped* **do**
4     $\mathcal{D}' \leftarrow B$ random samples from $\mathcal{D}$
5     flow $\leftarrow \sum_{\boldsymbol{x} \in \mathcal{D}'}$ CircuitFlow($\mathcal{C}, \boldsymbol{\theta}, \boldsymbol{x}$)
6     **for** *sum unit n and its child c* **do**
7         $\theta_{c|n}^{new} \leftarrow$ flow$[n,c]/$flow$[n]$
8         $\theta_{c|n} \leftarrow \alpha\theta_{c|n}^{(new)} + (1-\alpha)\theta_{c|n}$
---

# B Proofs

In this section, we provide detailed proofs of Theorem 1 (Section B.1) and Theorem 2 (Section B.2).

## B.1 Pruning One Edge over One Example

**Lemma 1** (Pruning One Edge Log-Likelihood Lower Bound). *For a PC $\mathcal{C}$ and a sample $\boldsymbol{x}$, the loss of log-likelihood by pruning away edge $(n, c)$ is*

$$\Delta\mathcal{LL}(\{\boldsymbol{x}\}, \mathcal{C}, \{(n,c)\}) = \log\left(\frac{1-\theta_{c|n}}{1-\theta_{c|n}+\theta_{c|n}\,\text{F}_n(\boldsymbol{x})-\text{F}_{n,c}(\boldsymbol{x})}\right) \leq -\log(1-\text{F}_{n,c}(\boldsymbol{x})).$$

*Proof.* For notation simplicit, denote the probability of units $m$ (resp. $n$) in the original (resp. pruned) PC given sample $\boldsymbol{x}$ as $p_m(\boldsymbol{x})$ (resp. $p'_n(\boldsymbol{x})$). As a slight extension of Definition 4, we define $F_n(\boldsymbol{x}; m)$ as the flow of unit $n$ w.r.t. the PC rooted at $m$.

The proof proceeds by induction over the PC's root unit. That is, we first consider pruning $(n, c)$ w.r.t. the PC rooted at $n$. Then, in the induction step, we prove that if the lemma holds for PC rooted at $m$, then it also holds for PC rooted at any parent unit of $m$. Instead of directly proving the statement in Lemma 1, we first prove that for any root node $m$, the following holds:

$$p_m(\boldsymbol{x}) - p'_m(\boldsymbol{x}) = F_n(\boldsymbol{x}; m) \cdot p_m(\boldsymbol{x}) \cdot \left(\frac{1}{1-\theta}\frac{\text{F}_{n,c}(\boldsymbol{x}; m)}{\text{F}_n(\boldsymbol{x}; m)} - \frac{\theta}{1-\theta}\right). \tag{4}$$

Base case: pruning an edge of the root unit. That is, the root unit of the PC is $n$. In this case, we have

$$p_n(\boldsymbol{x}) - p'_n(\boldsymbol{x}) = \sum_{c' \in \text{in}(n)} \theta_{c'|n} \cdot p_c(\boldsymbol{x}) - \sum_{c' \in \text{in}(n) \backslash c} \theta'_{c'|n} \cdot p'_c(\boldsymbol{x})$$

$$= \theta_{c|n} \cdot p_c(\boldsymbol{x}) + \sum_{c' \in \text{in}(n) \backslash c} \theta_{c'|n} \cdot p_c(\boldsymbol{x}) - \sum_{c' \in \text{in}(n) \backslash c} \theta'_{c'|n} \cdot p_c(\boldsymbol{x}), \qquad (5)$$

where $\theta'_{c|n}$ denotes the normalized parameter corresponding to edge $(n, c)$ in the pruned PC. Specifically, we have

$$\forall m \in \text{in}(n) \backslash c, \quad \theta'_{m|n} = \frac{\theta_{m|n}}{\sum_{c' \in \text{in}(n) \backslash c} \theta_{c'|n}} = \frac{\theta_{m|n}}{1 - \theta_{c|n}}.$$

For notation simplicity, denote $\theta := \theta_{c|n}$. Plug in the above definition into Equation 5, we have

$$p_n(\boldsymbol{x}) - p'_n(\boldsymbol{x}) = \theta_{c|n} \cdot p_c(\boldsymbol{x}) + \sum_{c' \in \text{in}(n) \backslash c} \theta_{c'|n} \cdot p_c(\boldsymbol{x}) - \frac{1}{1-\theta} \sum_{c' \in \text{in}(n) \backslash c} \theta_{c'|n} \cdot p_c(\boldsymbol{x})$$

$$= \theta_{c|n} \cdot p_c(\boldsymbol{x}) - \frac{\theta}{1-\theta} \sum_{c' \in \text{in}(n) \backslash c} \theta_{c'|n} \cdot p_c(\boldsymbol{x})$$

$$= \theta_{c|n} \cdot p_c(\boldsymbol{x}) - \frac{\theta}{1-\theta}(p_n(\boldsymbol{x}) - \theta_{c|n} p_c(\boldsymbol{x}))$$

$$= \frac{1}{1-\theta} \cdot \theta_{c|n} \cdot p_c(\boldsymbol{x}) - \frac{\theta}{1-\theta} \cdot p_n(\boldsymbol{x})$$

$$\overset{(a)}{=} \frac{1}{1-\theta} \cdot p_n(\boldsymbol{x}) \cdot \frac{F_{n,c}(\boldsymbol{x}; n)}{F_n(\boldsymbol{x}; n)} - \frac{\theta}{1-\theta} \cdot p_n(\boldsymbol{x})$$

$$= F_n(\boldsymbol{x}; n) \cdot p_n(\boldsymbol{x}) \cdot \left( \frac{1}{1-\theta} \frac{F_{n,c}(\boldsymbol{x}; n)}{F_n(\boldsymbol{x}; n)} - \frac{\theta}{1-\theta} \right), \qquad (6)$$

where $(a)$ follows from the fact that $F_n(\boldsymbol{x}; n) = 1$ and $F_{n,c}(\boldsymbol{x}; n) = \theta_{c|n} p_c(\boldsymbol{x})/p_n(\boldsymbol{x})$.

Inductive case #1: suppose Equation 4 holds for $m$. If product unit $d$ is a parent of $m$, we show that Equation 4 also holds for $d$:

$$p_d(\boldsymbol{x}) - p'_d(\boldsymbol{x}) = \prod_{n' \in \text{in}(d)} p_{n'}(\boldsymbol{x}) - \prod_{n' \in \text{in}(d)} p'_{n'}(\boldsymbol{x})$$

$$= (p_m(\boldsymbol{x}) - p'_m(\boldsymbol{x})) \prod_{n' \in \text{in}(d) \backslash m} p_{n'}(\boldsymbol{x})$$

$$\overset{(a)}{=} F_n(\boldsymbol{x}; m) \cdot p_m(\boldsymbol{x}) \cdot \left( \frac{1}{1-\theta} \frac{F_{n,c}(\boldsymbol{x}; m)}{F_n(\boldsymbol{x}; m)} - \frac{\theta}{1-\theta} \right) \cdot \prod_{n' \in \text{in}(d) \backslash m} p_{n'}(\boldsymbol{x})$$

$$\overset{(b)}{=} F_n(\boldsymbol{x}; d) \cdot p_d(\boldsymbol{x}) \cdot \left( \frac{1}{1-\theta} \frac{F_{n,c}(\boldsymbol{x}; d)}{F_n(\boldsymbol{x}; d)} - \frac{\theta}{1-\theta} \right),$$

where $(a)$ is the inductive step that applies Equation 6; $(b)$ follows from the fact that (note that $d$ is a product unit) $F_n(\boldsymbol{x}; m) = F_n(\boldsymbol{x}; d)$ and $F_{n,c}(\boldsymbol{x}; m) = F_{n,c}(\boldsymbol{x}; d)$.

Inductive case #2: for sum unit $d$, suppose Equation 4 holds for $m$, where $m \in \mathcal{A}$ iff $m \in \text{in}(d)$ and $m$ is an ancester of $n$ and $c$. Assume all other children of $d$ are not ancestoer of $n$, we show that

Equation 4 also holds for $d$:

$$p_d(\boldsymbol{x}) - p'_d(\boldsymbol{x}) = \theta_{m|d} \cdot (p_m(\boldsymbol{x}) - p'_m(\boldsymbol{x}))$$

$$= \theta_{m|d} \cdot F_n(\boldsymbol{x}; m) \cdot p_m(\boldsymbol{x}) \cdot \left( \frac{1}{1-\theta} \frac{F_{n,c}(\boldsymbol{x}; m)}{F_n(\boldsymbol{x}; m)} - \frac{\theta}{1-\theta} \right)$$

$$= \theta_{m|d} \cdot F_n(\boldsymbol{x}; m) \cdot p_m(\boldsymbol{x}) \cdot \left( \frac{1}{1-\theta} \frac{F_{n,c}(\boldsymbol{x}; d)}{F_n(\boldsymbol{x}; d)} - \frac{\theta}{1-\theta} \right)$$

$$= \theta_{m|d} \cdot F_n(\boldsymbol{x}; d) \cdot \frac{\sum_{m' \in \mathrm{in}(d)} \theta_{m'|d} p_{m'}(\boldsymbol{x})}{\theta_{m|d} p_m(\boldsymbol{x})} \cdot p_m(\boldsymbol{x}) \cdot \left( \frac{1}{1-\theta} \frac{F_{n,c}(\boldsymbol{x}; d)}{F_n(\boldsymbol{x}; d)} - \frac{\theta}{1-\theta} \right)$$

$$= F_n(\boldsymbol{x}; d) \cdot \left( \sum_{m' \in \mathrm{in}(d)} \theta_{m'|d} p_{m'}(\boldsymbol{x}) \right) \cdot \left( \frac{1}{1-\theta} \frac{F_{n,c}(\boldsymbol{x}; d)}{F_n(\boldsymbol{x}; d)} - \frac{\theta}{1-\theta} \right)$$

$$= F_n(\boldsymbol{x}; d) \cdot p_d(\boldsymbol{x}) \cdot \left( \frac{1}{1-\theta} \frac{F_{n,c}(\boldsymbol{x}; d)}{F_n(\boldsymbol{x}; d)} - \frac{\theta}{1-\theta} \right).$$

Therefore, following Equation 4 for root $r$, we have

$$\frac{p_r(\boldsymbol{x}) - p'_r(\boldsymbol{x})}{p_r(\boldsymbol{x})} = \frac{1}{1-\theta} F_{n,c}(\boldsymbol{x}; r) - \frac{\theta}{1-\theta} F_n(\boldsymbol{x}; r)$$

$$\Leftrightarrow \quad \frac{p'_r(\boldsymbol{x})}{p_r(\boldsymbol{x})} = 1 + \frac{\theta}{1-\theta} F_n(\boldsymbol{x}; r) - \frac{1}{1-\theta} F_{n,c}(\boldsymbol{x}; r)$$

Therefore, we have

$$\Delta \mathcal{LL}(\{\boldsymbol{x}\}, \mathcal{C}, \{(n,c)\}) = \log p_r(\boldsymbol{x}) - \log p'_r(\boldsymbol{x})$$

$$= \frac{1}{|\mathcal{D}|} \sum_{\boldsymbol{x} \in \mathcal{D}} \log \left( \frac{1 - \theta_{c|n}}{1 - \theta_{c|n} + \theta_{c|n} F_n(\boldsymbol{x}; r) - F_{n,c}(\boldsymbol{x}; r)} \right)$$

$$\overset{(a)}{\leq} -\log(1 - F_{n,c}(\boldsymbol{x})),$$

where $(a)$ follows from the fact that $F_{n,c}(\boldsymbol{x}) \leq F_n(\boldsymbol{x})$. $\qquad \square$

Theorem 1 follows directly from Lemma 1 by noting that for any dataset $\mathcal{D}$, $\Delta \mathcal{LL}(\mathcal{D}, \mathcal{C}, \{(n,c)\}) = \frac{1}{|\mathcal{D}|} \Delta \mathcal{LL}(\{\boldsymbol{x}\}, \mathcal{C}, \{(n,c)\})$.

### B.2 Pruning Multiple Edges

*Proof.* Similar to the proof of Lemma 1, we prove Theorem 2 by induction. Different from Lemma 1, we induce a slightly different objective:

$$p_m(\boldsymbol{x}) - p'_m(\boldsymbol{x}) \leq \sum_{(n,c) \in \mathcal{E} \cap \mathsf{des}(m)} F_n(\boldsymbol{x}; m) \cdot p_m(\boldsymbol{x}) \cdot \left( \frac{1}{1-\theta_{c|n}} \frac{F_{n,c}(\boldsymbol{x}; m)}{F_n(\boldsymbol{x}; m)} - \frac{\theta_{c|n}}{1-\theta_{c|n}} \right), \quad (7)$$

where $\mathsf{des}(n)$ is the set of descendent units of $n$.

Base case: the base case follows directly from the proof of Lemma 1, and lead to the conclusion in Equation 6.

Inductive case #1: suppose for all children of a product unit $d$, Equation 7 holds, we show that Equation 7 also holds for $d$:

$$p_d(\boldsymbol{x}) - p'_d(\boldsymbol{x}) = \prod_{m \in \mathsf{in}(d)} p_m(\boldsymbol{x}) - \prod_{m \in \mathsf{in}(d)} p'_m(\boldsymbol{x})$$

$$= \prod_{m \in \mathsf{in}(d)} p_m(\boldsymbol{x}) - \prod_{m \in \mathsf{in}(d)} \Big( p_m(\boldsymbol{x}) - (p_m(\boldsymbol{x}) - p'_m(\boldsymbol{x})) \Big)$$

$$\leq \sum_{m \in \mathsf{in}(d)} \Big( p_m(\boldsymbol{x}) - p'_m(\boldsymbol{x}) \Big) \cdot \prod_{m' \in \mathsf{in}(d) \setminus m} p_{m'}(\boldsymbol{x})$$

$$\overset{(a)}{\leq} \sum_{m \in \mathsf{in}(d)} \sum_{(n,c) \in \mathcal{E} \cap \mathsf{des}(m)} F_n(\boldsymbol{x}; d) \cdot p_d(\boldsymbol{x}) \cdot \left( \frac{1}{1 - \theta_{c|n}} \frac{F_{n,c}(\boldsymbol{x}; m)}{F_n(\boldsymbol{x}; m)} - \frac{\theta_{c|n}}{1 - \theta_{c|n}} \right)$$

$$\leq \sum_{(n,c) \in \mathcal{E} \cap \mathsf{des}(d)} F_n(\boldsymbol{x}; d) \cdot p_d(\boldsymbol{x}) \cdot \left( \frac{1}{1 - \theta_{c|n}} \frac{F_{n,c}(\boldsymbol{x}; d)}{F_n(\boldsymbol{x}; d)} - \frac{\theta_{c|n}}{1 - \theta_{c|n}} \right),$$

where $(a)$ uses the definition that $p_d(\boldsymbol{x}) = \prod_{m \in \mathsf{in}(d)} p_m(\boldsymbol{x})$.

Inductive case #2: suppose for all children of a sum unit $d$, Equation 7 holds, we show that Equation 7 also holds for $d$:

$$p_d(\boldsymbol{x}) - p'_d(\boldsymbol{x}) = \sum_{m \in \mathsf{in}(d) \cap (d,m) \notin \mathcal{E}} \theta_{m|d} \cdot \Big( p_m(\boldsymbol{x}) - p'_m(\boldsymbol{x}) \Big) + \sum_{m \in \mathsf{in}(d) \cap (d,m) \in \mathcal{E}} \theta_{m|d} \cdot \Big( p_m(\boldsymbol{x}) - p'_m(\boldsymbol{x}) \Big)$$

$$\overset{(a)}{=} \sum_{m \in \mathsf{in}(d) \cap (d,m) \notin \mathcal{E}} \theta_{m|d} \cdot \Big( p_m(\boldsymbol{x}) - p'_m(\boldsymbol{x}) \Big)$$

$$+ \sum_{m \in \mathsf{in}(d) \cap (d,m) \in \mathcal{E}} \theta_{m|d} \cdot F_n(\boldsymbol{x}; m) \cdot p_m(\boldsymbol{x}) \cdot \left( \frac{1}{1 - \theta_{c|n}} \frac{F_{n,c}(\boldsymbol{x}; m)}{F_n(\boldsymbol{x}; m)} - \frac{\theta_{c|n}}{1 - \theta_{c|n}} \right),$$

where $(a)$ follows from the base case of the induction. Next, we focus on the first term of the above equation:

$$\sum_{m \in \mathsf{in}(d) \cap (d,m) \notin \mathcal{E}} \theta_{m|d} \cdot \Big( p_m(\boldsymbol{x}) - p'_m(\boldsymbol{x}) \Big)$$

$$\leq \sum_{m \in \mathsf{in}(d) \cap (d,m) \notin \mathcal{E}} \sum_{(n,c) \in \mathcal{E} \cap \mathsf{des}(m)} \theta_{m|d} \cdot \Big( p_m(\boldsymbol{x}) - p'_m(\boldsymbol{x}) \Big)$$

$$\leq \sum_{m \in \mathsf{in}(d) \cap (d,m) \notin \mathcal{E}} \sum_{(n,c) \in \mathcal{E} \cap \mathsf{des}(m)} \theta_{m|d} \cdot F_n(\boldsymbol{x}; m) \cdot p_m(\boldsymbol{x}) \cdot \left( \frac{1}{1 - \theta_{c|n}} \frac{F_{n,c}(\boldsymbol{x}; m)}{F_n(\boldsymbol{x}; m)} - \frac{\theta_{c|n}}{1 - \theta_{c|n}} \right)$$

$$\leq \sum_{(n,c) \in \mathcal{E} \cap \mathsf{des}(d)} F_n(\boldsymbol{x}; d) \cdot p_d(\boldsymbol{x}) \cdot \left( \frac{1}{1 - \theta_{c|n}} \frac{F_{n,c}(\boldsymbol{x}; d)}{F_n(\boldsymbol{x}; d)} - \frac{\theta_{c|n}}{1 - \theta_{c|n}} \right),$$

where the derivation of the last inequality follows from the corresponding steps in the proof of Lemma 1.

Therefore, from Equation 7, we can conclude that

$$\Delta \mathcal{LL}(\mathcal{D}, \mathcal{C}, \mathcal{E}) \leq -\frac{1}{|\mathcal{D}|} \sum_{\boldsymbol{x}} \log\Big(1 - \sum_{(n,c) \in \mathcal{E}} F_{n,c}(\boldsymbol{x})\Big).$$

Finally, we prove the approximation step in Equation 3. Let $\epsilon(\cdot) = \sum_{(n,c) \in \mathcal{E}} F_{n,c}(\cdot) \in [0,1)$. We have,

$$\text{RHS} = -\sum_{\boldsymbol{x} \in \mathcal{D}} \log(1 - \epsilon(\boldsymbol{x})) = -\sum_{\boldsymbol{x} \in \mathcal{D}} \sum_{k=1}^{\infty} -\frac{\epsilon(\boldsymbol{x})^k}{k} \text{(Taylor expansion)} \leq \sum_{\boldsymbol{x} \in \mathcal{D}} \sum_{k=1}^{\infty} \epsilon(\boldsymbol{x})^k$$

$$= \sum_{\boldsymbol{x} \in \mathcal{D}} \frac{\epsilon(\boldsymbol{x})}{1 - \epsilon(\boldsymbol{x})} = \frac{1}{1 - \epsilon} \sum_{\boldsymbol{x} \in \mathcal{D}} \epsilon(\boldsymbol{x}) = \frac{1}{1 - \epsilon} \sum_{(n,c) \in \mathcal{E}} \sum_{\boldsymbol{x} \in \mathcal{D}} F_{n,c}(\boldsymbol{x}) = \frac{1}{1 - \epsilon} \sum_{(n,c) \in \mathcal{E}} F_{n,c}(\mathcal{D}).$$

$\square$

# C    Experiments Details

**Hardware specifications**    All experiments are performed on a server with 32 CPUs, 126G Memory, and NVIDIA RTX A5000 GPUs with 26G Memory. In all experiments, we only use a single GPU on the server.

## C.1    Datasets

For MNIST-family datasets, we split 5% of training set as validation set for early stopping. For Penn Tree Bank dataset, we follow the setting in Mikolov et al. [29] to split a training, validation, and test set. Table 3 lists the all the dataset statistics.

Table 3: Dataset statistics including number of variables (**#vars**), number of categories for each variable (**#cat**), and number of samples for training, validation and test set (**#train**, **#valid**, **#test**).

| Dataset | $n$ (**#vars**) | $k$ (**#cat**) | **#train** | **#valid** | **#test** |
|---|---|---|---|---|---|
| MNIST | 28×28 | 256 | 57000 | 3000 | 10000 |
| EMNIST(MNIST) | 28×28 | 256 | 57000 | 3000 | 10000 |
| EMNIST(Letters) | 28×28 | 256 | 118560 | 6240 | 20800 |
| EMNIST(Balanced) | 28×28 | 256 | 107160 | 5640 | 18800 |
| EMNIST(ByClass) | 28×28 | 256 | 663035 | 34897 | 116323 |
| FashionMNIST | 28×28 | 256 | 57000 | 3000 | 10000 |
| Penn Tree Bank | 288 | 50 | 42068 | 3370 | 3761 |

## C.2    Learning Hidden Chow-Liu Trees

**HCLT structures.**    Adopting hidden chow liu tree (HCLT) PC architecture as in Liu and Van den Broeck [25], we reimplement the learning process to speed it up and use a different training pipeline and hyper-parameters tuning.

**EM parameter learning**    We adopt the EM parameter learning algorithm introduced in Choi et al. [4], which computes the EM update target parameters using circuit flows. We use a stochastic mini-batches EM algorithm. Denoting $\theta^{\texttt{new}}$ as the EM update target computed from a mini-batch of samples, and we update the targeting parameter with a learning rate $\alpha$: $\theta^{t+1} \leftarrow \alpha\theta^{\texttt{new}} + (1 - \alpha)\theta^t$. $\alpha$ is piecewise-linearly annealed from $[1.0, 0.1]$, $[0.1, 0.01]$, $[0.01, 0.001]$, and each piece is trained $T$ epochs.

**Hyper-parameters searching.**    For all the experiments, the hyper-parameters are searched from

- $h \in \{8, 16, 32, 64, 128, 256\}$, the hidden size of HCLT structures;

- $\gamma \in \{0.0001, 0.001, 0.01, 0.1, 1.0\}$, Laplace smoothing factor;

- $B \in \{128, 256, 512, 1024\}$, batch-size in mini-batches EM algorithm;

- $\alpha$ piecewise-linearly annealed from $[1.0, 0.1]$, $[0.1, 0.01]$, $[0.01, 0.001]$, where each piece is called one mini-batch EM phase. Usually the algorithm will start to overfit as validation set and stop at the third phase;

- $T = 100$, number of epochs for each mini-batch EM phase.

The PC size is quadratically growing with hidden size $h$, thus it is inefficient to do a grid search among the entire hyper-parameters space. What we do is to fist do a grid search when $h = 8$ or $h = 16$ to find the best Laplace smoothing factor $\gamma$ and batch-size $B$ for each dataset, and then fix $\gamma$ and $B$ to train a PC with larger hidden size $h \in \{32, 64, 128, 256\}$. The best tuned $B$ is in $\{256, 512\}$, which is different for different hidden size $h$, and the best tuned $\gamma$ is 0.01.

## C.3 Details of Section 6.1

**Sparse PC (ours).** Given an HCLT learned in Section C.2 as initial PC, we use the structure learning process proposed in Section 5. Specifically, starts from initial HCLT, for each iteration, we (1) prune 75% of the PC parameters, and (2) grow PC size with Gaussian variance $\epsilon$, (3) finetuing PC using mini-batches EM parameter learning with learning rate $\alpha$. We prune and grow PC iteratively until the validation set likelihood is overfitted . The hyper-parameters are searched from

- $\epsilon \in \{0.1, 0.3, 0.5\}$, Gaussian variance in growing operation;
- $\alpha$, piecewise-linearly annealed from $[0.1, 0.01]$, $[0.01, 0.001]$;
- $T = 50$, number of epochs for each mini-batch EM phase;
- for $\gamma$ and $B$, we use the tuned best number from Section C.2.

**HCLT.** The HLCT experiments in Table 1 are performed following the original paper (Code `https://github.com/UCLA-StarAI/Tractable-PC-Regularization`), which is different from the leaning pipeline we use as our inital PC (Section C.2).

**SPN.** We reimplement the SPN architecture ourselves following Peharz et al. [34] and train it with the same mini-batch pipeline as HCLT.

**IDF.** We run all experiments with the code in the GitHub repo provided by the authors. We adopt an IDF model with the following hyperparameters: 8 flow layers per level; 2 levels; densenets with depth 6 and 512 channels; base learning rate 0.001; learning rate decay 0.999. The algorithm adopts an CPU-based entropy coder rANS.

**BitSwap.** We train all models using the following author-provided script: `https://github.com/fhkingma/bitswap/blob/master/model/mnist_train`.

**BB-ANS.** All experiments are performed using the following official code `https://github.com/bits-back/bits-back`.

**McBits.** All experiments are performed using the following official code `https://github.com/ryoungj/mcbits`.

## C.4 Details of Section 6.2

For all experiments in Section 6.2, we use the best tuned $\gamma$ and $B$ from Section C.2 and hidden size $h$ ranging from $\{16, 32, 64, 128\}$. For experiments "What is the Smallest PC for the Same Likelihood?", the hyper-parameters are searched from

- $k \in \{0.05, 0.1, 0.3\}$, percentage of parameters to prune each iteration;
- $\alpha$, piecewise-linearly annealed from $[0.3, 0.1]$, $[0.1, 0.01]$, $[0.01, 0.001]$;
- $T = 50$, number of epochs for each mini-batch EM phase;

For experiments "What is the Best PC Given the Same Size?", we use the same setting as in Section C.3.