# OpenReview forum: "Sparse Probabilistic Circuits via Pruning and Growing"
_NeurIPS.cc/2022/Conference — NeurIPS 2022 Accept_

### Official Review · Reviewer_BazA · 2022-06-24

**Rating:** 7
**Confidence:** 4
**Soundness:** 4 excellent
**Presentation:** 4 excellent
**Contribution:** 3 good

**Summary:**

This work addresses the scalability issue of the probabilistic circuits. They propose a technique of pruning and growing to scale and improve the expressiveness of the PCs. For pruning, they propose pruning edges based on the probability that it will be `activated’ during sampling (eFlow strategy in Alg.1). To counter the reduction in the model capacity done by the pruning step, they introduce a Grow (Alg.2) strategy. To obtain a compressed PC, they alternate between the pruning & growing operations based on optimizing for the log-likelihood.

**Questions:**

Questions &  Suggestions:
1. I found the definition of root node confusing initially. Usually, a DAG will have sources and sinks. I understand that if the nomenclature followed in the previous works states the same but if not, please update the naming.
2. (Line 105) thee -> the
3. The datasets used in this paper are still small in size IMO. Will be helpful to see the scaling on larger datasets. Alternatively, if the authors can report the limitations of their approach for larger datasets, that will be appreciated as well.
4. What is the intuition in using the hidden Chow-Liu Tree as the initial PC structure for the experiments?
5. (Viewing as a structure learning algorithm for the PC) I suspect that after compressing the initial PC structure, one would end up with a larger (due to the grow operation) but a sparser (due to prune) structure. Then the choice of initial PC structure will play a key role. The grow operation will increase the params by a factor of 4, so please share your thoughts on using 75% pruning?
6. How many iterations (between grow + prune) are needed for convergence for the MNIST and PTB tasks?


**Limitations:**

As mentioned above, will be good to know the scale of the data this method can currently handle.

**Strengths And Weaknesses:**

Pros & Cons:
1. It is a well-written paper. I appreciate the authors introducing the literature and prior work about PCs in a concise and efficient manner.
2. The formulation of pruning by circuit flows is a natural follow-up question to the pruning by generative significance. The sampled conditioned parameterization is one approach which the authors introduced in this work. This may introduce a bias/overfitting for the training data? I wonder if there are other approaches that can better handle the same problem. What if we do unconditional sampling and then measure the probability of observing the sampled data with the observed data. Use this information to weigh the flow?

This is good work, although I feel the scalability performance is still far from using it on larger datasets (at least from the experiments added in the paper).

---

> ### Author Response · Authors · 2022-08-02
> **Author Response**
>
> We thank the reviewer for their time spent reading and reviewing our paper.
>
> > this may introduce a bias/overfitting for the training data
>
> Indeed this may introduce overfitting for the training data, but empirically the current circuit flows method works at the current scale. If overfitting would become an issue in other scenarios, one can split out a validation set to compute circuit flows from or prune according to generative significance.
>
> [**Question 2**] In PC literature, people usually refer to inputs and outputs. We will add “sinks” and “sources” to point to the DAG literature in the revised version.
>
> [**Question 3**]  Despite significant recent progress in scaling up PC learning to larger datasets (full MNIST was out of reach until even a year ago), we agree that the proposed PC learning algorithm still has limitations on scaling to larger natural image datasets like `Cifar10`. We expected the increased scaling of PC learners to continue in future.
>
> [**Question 4**] The pruning and growing algorithm can be applicable to all PC structures. We use HCLT as the initial structure since it is the current state-of-the-art structure. HCLT puts variables with high correlations close together, which makes it a good density estimator.
>
> [**Question 5**] In one of the experiments setting, we keep the number of parameters to be the same as the initial structure after pruning and growing, and since the growing operation increases the number of parameters by 4x, we prune 75% of the parameters so the resulting circuit has the same number of parameters as the initial circuit. Pruning fewer parameters such that after pruning and growing, the model size gradually increases is interesting future work.
>
> [**Question 6**] Usually ~100 EM epochs to finetune parameters after pruning and growing, and ~5 pruning and growing phases.

---

> > ### Comment · Reviewer_BazA · 2022-08-07
> > **[Response to authors]**
> >
> > Thank you for following up on my questions. This is good work and I hope to see the research community take it forward.

---

### Official Review · Reviewer_QUWk · 2022-07-02

**Rating:** 9
**Confidence:** 4
**Soundness:** 4 excellent
**Presentation:** 4 excellent
**Contribution:** 4 excellent

**Summary:**

The authors present an algorithm to distill probabilistic circuits (PCs). In this particular case, they present pruning and growing operations, removing nodes that are not significant for the estimator and adding nodes where needed.

The authors present and prove multiple variants for pruning, describe structure learning via pruning and growing and parameter estimation via stochastic mini-batch EM.
Furthermore, the strong empirical results indicate that this approach can have significant impact in the learning of PCs.

This approach also allows practitioners to control between model size and performance.



**Questions:**

Although this paper proposes a very strong approach as shown by the empirical results. There are some important aspects open.

I'm wondering about the impact of the pruning and growing operations with respect to the impact they have in PCs on their performance as density estimators.

Consider starting with a large trained PC. Is the removal of a particular node via your pruning, something that increases the likelihood? Or are those extra nodes simply overlapping other parts of the distribution but have no negative impact (other than size) by being there? In 6.2 you mention that it is better than starting from a smaller size, but you do not mention what happens if you start big, but prune and optimize without growing.

Is the improvement in the performance all due to the growing operation? I could understand that if the model lacks capacity in a particular node then it would be very beneficial to extend there. But is that the operation responsible for the improvement?
Or is the improvement coming from the noise injected by the growing operation? This could potentially push the optimizer into regions it hasn't explored well before.

In short, more insights or experiments on what is causing the improvement would make the paper even stronger.

**Limitations:**

no negative societal impact.

**Strengths And Weaknesses:**

The paper presents a solid contribution as shown by the empirical evaluation, pushing state-of-the-art in PCs.

The paper is well-written and technically sound.

There are some minor weaknesses:
In [1] the authors propose a pruning approach based on the lottery ticket hypothesis for PCs, I believe this might be relevant here.
There is no comparison to other methods that work on regularization of PCs such as [2].

[1] Ventola, Fabrizio, et al. "Residual Sum-Product Networks." International Conference on Probabilistic Graphical Models. PMLR, 2020.
[2] Shih, Andy, Dorsa Sadigh, and Stefano Ermon. "HyperSPNs: compact and expressive probabilistic circuits." Advances in Neural Information Processing Systems 34 (2021): 8571-8582.

---

> ### Author Response · Authors · 2022-08-02
> **Author Response**
>
> We thank the author for their time spent reading and reviewing our work.
>
> [**Strengths and Weakness**] We thank the reviewer for pointing out relevant work on PC pruning and regularization. We have added discussions in the revised manuscript.
>
> > but you do not mention what happens if you start big, but prune and optimize without growing.
>
> Thank you for the insightful comment. Based on our experiments, the training performance of a large circuit easily plateaus as we increase the model size. Therefore, training very large PCs from scratch faces optimization challenges. Hence, starting with a big PC but pruning and optimizing without growing does not provide too much performance gain. In contrast, iterative pruning and growing leads to better performance since the structure after growing provides a good initial point for the parameter optimizer.
>
>
>
> > Is the improvement in the performance all due to the growing operation?  Or is the improvement coming from the noise injected by the growing operation?
>
> The benefits come from both sides: increasing the capacity of certain nodes and injecting noise. The growing operation increases the capacity, and the noise injected helps the optimizer redistribute the samples and find another set of parameters for newly created nodes. If the nodes do not have enough capacity, the growing operation increases it; if the nodes have enough capacity, the pruning operation afterward will remove the unnecessary capacity. However, we discover that the training performance plateaus after we increase the mode size, which means if given a model with large enough capacity, the optimizer is not working, and it can not find a good solution since the parameter space is too complex. Therefore we need the growing operation. Intuitively, the optimizer finds a good solution in lower dimension (smaller hidden size), after growing, the model capacity is increased and the optimizer finds a better solution starting from the ones in lower dimension.

---

### Official Review · Reviewer_Y5Dn · 2022-07-12

**Rating:** 7
**Confidence:** 3
**Soundness:** 3 good
**Presentation:** 4 excellent
**Contribution:** 4 excellent

**Summary:**

This work suggests a method for exploiting sparsity in probabilistic circuits (PCs). The authors motivate the paper by showing PC parameters being close to zero in the majority of PCs' sum edges. Thus, these models could be reduced with a potentially negligible loss in representation (log-likelihood). The paper suggests two methods for compressing a PC while minimizing its representation loss: pruning and growing. Pruning identifies and removes parts of the PC that can be shown as non-important under the PC-represented distribution. Similarly, growing mitigates representation loss by adding new edges with noisy parameters, which can be optimized to achieve better performance. Applying "pruning and growing" can significantly reduce the model's size while not losing much performance.

**Questions:**

* It would be interesting to investigate if any of these relevance metrics could be applicable in other models, such as the competitors in Table 1 since the metrics are either magnitude or sampling-based.
* In the literature, PC architectures with intractable parts can be found either at the leaf level or internal nodes. Could the metrics suggested here be adapted or used in such architectures?
* Although the work is well-motivated, a suggestion for improvement would be to relate this work's results to real-world applications and previous literature practical results. For instance, the compression gains with minor likelihood loss could be associated with specific application scenarios. Moreover, referencing previous works' struggles with model size might further ground the paper's results.

**Limitations:**

This paper does not explicitly discuss its limitations. Since the experimental results are a crucial draw for this work, it could be beneficial to add a short discussion on how the metrics used to measure exploiting sparsity benefit or hurts the different models being evaluated.

**Strengths And Weaknesses:**

Some of this work's strengths are its comprehensive solution and clear, practical relevance. Regarding the comprehensive solution, this paper puts forth the problem of computation sparsity in PCs. This problem involves both tasks of removing unnecessary parts of this computation while minimizing loss in performance. The solution suggested in this manuscript tackles both tasks by providing modular techniques called pruning and growing. These techniques allow for fine-tuning the gains and losses of compression and log-likelihood, respectively, and they are individually grounded on theoretical results. For instance, the pruning technique called eFlow is a good approximation for the upper bound of the log-likelihood drop when removing edges, as shown in Theorem 1 and 2. These comprehensive discussions of the sparsity problem, including theoretical analysis, strengthen the work here.
This paper's practical relevance is evident in its motivation and provides state-of-the-art results on benchmark experiments. Indeed, the manuscript emphasizes the impact of sparsity in the model size, as depicted in Figure 1, while also showing methods for maximizing efficiency, as discussed in Section 5. Thus, the work being suggested here is relevant for practical applications. This premise is confirmed with state-of-the-art results in challenging datasets. Furthermore, the extended discussion in Section 6.2 is exciting, and it shows the capabilities of pruning-growing operations on fine-tuning the trade-off between PC size and accuracy.

One minor weakness in this paper's presentation lies in the intuition behind the "theoretically-grounded metric" for the importance of PC edges. The manuscript says that the "global influence of edges on the PC's output" is "highly related to PC semantics." This point is positive for the use of PCs compared to other models, such as some competitors in Table 1, since PC's semantics are a differentiator characteristic. However, theoretical discussions in Section 4 are focused on one pruning method, EFlow, which bases itself on PC's sampling process. Thus, it is unclear from the text how the theoretical analysis uses PC's unique semantics.

---

> ### Author Response · Authors · 2022-08-02
> **Author Response**
>
> We thank the reviewer for their time in reading and reviewing our work and evaluating it to be with high impact.
>
>
> > However, theoretical discussions in Section 4 are focused on one pruning method, EFlow, which bases itself on PC's sampling process.
>
> The theoretical discussion aims to find the best set of edges such that pruning this set minimizes the log-likelihood drop. And we discover that the eFlow is an approximation of the tight bound of the oracle set, while also being connected to the sampling semantics. PC’s unique semantics are used to prove the theorems, for example, the induction case in our proofs. Our theoretical analysis highlights the benefits of the “pruning by generative significance” idea.
>
>
> [**Question 1**] The proposed pruning and growing algorithms can be applied to arbitrary PCs, including HCLT and RatSPN. We use HCLT as our initial structure since it is a state-of-the-art structure for various datasets. The idea of pruning by circuit flows can not be directly applied to other neural network-based deep generative models since evaluating the importance of a single weight/edge to the model’s generative performance is hard. However, the high-level idea of pruning by the sampling measured statistics can be widely applicable to all types of models. This could be interesting future work.
>
> [**Question 2**] If the PC has intractable leafs, the flows are still well defined (albeit on top of possibly approximate leaf likelihoods) and our method should work as is. If inner nodes are intractable (that is, a non-decomposable product) more care needs to be taken; we did not explore this extension.
>
> [**Question 3**] The distilled PC can benefit applications that require efficient real time inferences such as missing value imputation and PC-based compression. Prior works on PC learners like RatSPN and HCLT usually aim for better likelihoods and rarely trade off the model size.
>
>
> [**Limitations**] We agree that showing the limitations is very important. The baseline models are mostly dense models, therefore we do not evaluate the sparsity per se. One limitation we mention in the paper is the fact that custom CUDA kernels are required to exploit the sparsity of the PC and reduce runtime. Of course the usual risks of ML evaluation apply as well. We will add a short discussion in the next version.

---

### Official Review · Reviewer_YPFA · 2022-07-12

**Rating:** 8
**Confidence:** 4
**Soundness:** 4 excellent
**Presentation:** 4 excellent
**Contribution:** 3 good

**Summary:**

An approach to pruning and growing probabilistic circuits is proposed, which is empirically shown to be useful for achieving higher effective utilization of the weights of the circuit and thus better likelihoods on various datasets.

**Questions:**

The approach of evaluating the probability of reaching some node of a probabilistic tree feels certain to have been used before in other settings. Can we find prior work that this would connect with?

Table 1: How many parameters are used by each of these models? How robust is the method (std devs of ~10 fits)? How long does each method take to fit?

Sec 6.2: The criteria of "log likelihood does not decrease by more that 1%" seems a little arbitrary, like a sliding scale. Is this commonly done in the model compression/pruning literature? I've seen results about changes in top-1/top-5 % accuracy after pruning. To put it differently: a change from 1.6 bpd to 1.7 bpd is more significant than a change from .16 to .17, and Fig 6 is comparing at least three different scales (1.2, 1.8, 3.4).

Sec 7: Can we say anything about model selection / Bayes information criterion for PCs? Does the pruning/growing approach give us a way of finding a BIC-optimal PC?

**Limitations:**

line 229 addresses a limitation - sparse PC is not vectorization-friendly and requires custom GPU kernels. These are provided in OSS.

No societal impact noted by authors or reviewer.

**Strengths And Weaknesses:**

Originality
The approach to pruning ends up being quite simple, evaluating the probability of sampling along any given branch of the circuit's tree and using that as an upper bound for the influence that portion of the tree can have on the overall joint density, then pruning out low-influence branches. I have not seem similar work in PCs, but would not be surprised to be able to find something related if looking at work in trees, graphs, etc. In fact, it would be nice if we could point to some work on decision trees or graph/network flow as algorithmically related work in this space. Relative to other approaches to learning PC structure, the prune and grow approach seems possibly simpler, and the empirical results suggest it is more robust.

Quality
The work is well done and experiments seem typical for the PCs + density estimators space (MNIST family + PTB). The overall construction of the paper is sound, approach is well explained.
The paper needs proofreading. Multiple misspellings, grammar issues (lines 30, 31, 38, 59, 87, 105, 198, 240, 279). Rather than "term" on lines 171, 172, 181, 185, perhaps "expression" would be a better word. Line 194, isn't it impossible to get a more expressive PC by pruning (not just "difficult")? Line 271, I would use "k portion" or "k fraction" instead of "k%" because k% makes it ambiguous whether k% of .1 means .1 fraction or .001 fraction.

Clarity
Presentation is clear, and the experiments demonstrate the value incremental to multiple competitive baselines. Presumably these results would also map to more effective lossless PC-based compression, which might have been a nice story to tag onto the experiments.

Significance
This work is important in the space of density estimation & probabilistic circuits, which historically has been of interest to a broad swath of conference participants. Achieves new SoTAs in this class for bpd. The technique is also relevant in the wider space of model pruning, which is also of broad interest.

---

> ### Author Response · Authors · 2022-08-02
> **Author Response**
>
> We thank the reviewer for their detailed comments and evaluating our work to be sound, well written, interesting, and with good quality. We address the issues and answer the questions as follows.
>
> [Originality]
> > but would not be surprised to be able to find something related if looking at work in trees, graphs, etc.
>
> Thank you for pointing out potentially relevant work from the tree/graph literature. There are works on pruning decision trees according to validation accuracy reduction, which is similar to the prune by generative significance approach proposed in the paper. We have added discussions in the revised manuscript.
>
> Patil D D, Wadhai V M, Gokhale J A. Evaluation of decision tree pruning algorithms for complexity and classification accuracy[J]. International Journal of Computer Applications, 2010, 11(2): 23-30.
>
> [Quality] We have revised the typos, and grammar issues in the revised manuscript. Thank you for pointing them out.
>
> > Line 194, isn't it impossible to get a more expressive PC by pruning (not just "difficult")?
>
> It is indeed impossible to increase the capacity/expressiveness by pruning. What we meant to say is that it is difficult to improve likelihood, which can happen by pruning over- or under-fitting subcircuits, but is limited and rare (it does happen doing little pruning on some large PCs).
>
> [Clarity]
>
> > these results would also map to more effective lossless PC-based compression
>
> Thanks for pointing out the connection between the proposed pruning/growing algorithm to PC-based compression algorithms. Since we can distill a smaller PC with comparable likelihoods, there is the potential of significantly speeding up PC-based compression. We will mention this connection in the introduction of the next version.
>
> [Table 1] We will add the full details about model size, training time, and std dev in the next version. In short, for MNIST with 128 hidden states, training takes around 8 hours, and the number of parameters (excluding leaf nodes) is around 8.4 M.
>
> [Sec 6.2] Indeed, it is always difficult to summarize such trade-offs in a single number. We choose the relative likelihood evaluation in Section 6.2 because the bpd of different datasets are in different ranges. We are happy to also report other metrics based on your suggestion.
>
> [Sec 7] The pruning and growing algorithms we propose take a more heuristic strategy to pick edges. We think in principle BIC can be very related to finding the best subsets of edges to prune/grow. We thank the reviewer for pointing it out, and we will leave it as promising future work.

---

### Meta-Review · Area_Chair_5oNN · 2022-08-22

**Recommendation:** Accept
**Confidence:** Certain

**Metareview:**

The paper introduces sparseness-inducing techniques for probabilistic circuits (PCs), leading to novel structure learning approaches for PCs. The reviewers were very positive about this paper, found it well-written and to be improving state-of-the-art. The techniques are novel and shown to be effective on generative modeling tasks.


**Award:**

No

---

### Decision · Program_Chairs · 2022-09-14

Accept